

# The fermionic double smeared null energy condition

**Duarte Fragoso[1⋆] and Lihan Guo[1,2†]**

**1** ITFA, Universiteit van Amsterdam, Science Park 904, 1098 XH Amsterdam, the Netherlands
**2** Dipartimento di Matematica e Fisica, Universit'a Cattolica del Sacro Cuore,
Via della Garzetta 48, 25133 Brescia, Italy

⋆ duarte.fragoso@student.uva.nl , † lihan.guo@unicatt.it

## Abstract

Energy conditions are crucial for understanding why exotic phenomena such as traversable wormholes and closed timelike curves remain elusive. In this paper, we prove the Double Smeared Null Energy Condition (DSNEC) for the fermionic free theory in 4-dimensional flat Minkowski spacetime, extending previous work on the same energy condition for the bosonic case [1, 2] by adapting Fewster and Mistry's method [3] to the energy-momentum tensor $T_{++}$. A notable difference from previous works lies in the presence of the $\gamma_0\gamma_+$ matrix in $T_{++}$, causing a loss of symmetry. This challenge is addressed by making use of its square-root matrix. We provide explicit analytic results for the massless case as well as numerical insights for the mass-dependence of the bound in the case of Gaussian smearing.

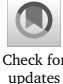

# 1   Introduction

In general relativity, Einstein's equation itself doesn't impose any restrictions on the form of the energy-momentum tensor $T_{\mu\nu}$. This freedom allows the existence of solutions $G_{\mu\nu}$ that may lead to surprising phenomena, such as macroscopic traversable wormholes [4], closed timelike curves [5] or other causality violations. Energy conditions are essential for explaining why these phenomena have never been observed. The Null Energy Condition (NEC) is particularly important since it is essential in the proof of Penrose's singularity theorem [6] and the second law of black hole thermodynamics (or the Area Theorem) [7] [8] [9].

Previous work has been done for new types of energy conditions namely the Smeared Null Energy Condition (SNEC) [10] and the Double-Smeared Null Energy Condition (DSNEC) [1] [2] which were used to deal with problems that arise when generalizing the NEC to a quantum setup [11–19]. Since this was only studied for the free bosonic theory, in this paper, we will focus on the SNEC and DSNEC for the fermionic theory. Our derivation will closely follow the reasoning of Fewster and Mistry [3] on the Quantum Weak Energy Inequalities for the Dirac field, who deduced a bound for the $T_{00}$ component of the massive fermionic free theory in four-dimensional flat Minkowski spacetime, and Wei-Wing et al [20], who generalized this result for Minkowski spacetime of arbitrary dimensions.

We introduce operators $\mathcal{O}_{\mu i}$ that enable us to express the smeared energy-momentum tensor as the difference between a positive semi-definite operator and a c-number. The primary challenge in defining these operators arises from the presence of the $\gamma_0\gamma_+$ matrix in $T_{++}$, which reduces the symmetry of the problem. This obstacle is overcome by incorporating the square-root matrix of $\gamma_0\gamma_+$ in the definition of $\mathcal{O}_{\mu i}$.

In brief, the structure of the paper is as follows: In Section 2, we undertake the derivation outlined above and obtain an inequality for the once-smeared $T_{++}$. However, this inequality is completely trivial, i.e. the lower bound obtained is $-\infty$. We address this issue in Section 3.1 by applying the smearing in two directions, providing a new energy condition:

$$\langle T_{f_+f_-}\rangle \geq -\frac{1}{4\pi^4}\int_0^\infty du \int_{\frac{m^2}{u}}^\infty dv \left(\frac{vu^3}{6} - \frac{m^2u^2}{2} + \frac{m^4u}{2v} - \frac{m^6}{6v^2}\right)|\hat{g}_+(u)|^2|\hat{g}_-(v)|^2, \quad (1)$$

where $f_\pm$ are smearing functions in spacetime coordinates and $\hat{g}_\pm$ denotes the Fourier transform of $\sqrt{f_\pm}$. Additionally, we present explicit results for the massless case in Section 3.2, where we employ a Gaussian distribution as the smearing function and derive a bound that depends rationally on the standard deviations,

$$\langle T_{f_+f_-}\rangle \geq -\frac{1}{96\pi^4\sigma_+^3\sigma_-}. \quad (2)$$

Finally, in Section 3.3, we provide numerical results concerning the mass-dependence of the bound. In particular, we observe that for large masses, the bound asymptotically tends to zero.

# 2   Derivation of the smearing null energy condition

In this section, we will derive a bound for the $T_{++}$ component of the energy-momentum tensor when smeared over the $x^+$-direction.[1] The quantum field theory considered is the free fermion in Minkowski flat spacetime. Note that, despite the bound derived being trivial, the idea can and will be used to deduce a non-trivial bound in Section 3.1.

---

[1]The light-cone variables $x^+$ and $x^-$ are defined in Appendix A, as well as the light-cone momentum coordinates $k^+$ and $k^-$.

First, let us write the symmetrized version of the energy-momentum tensor for the free fermion (the Belinfante tensor):

$$T_{\mu\nu} = \frac{i}{4}(\bar{\psi}\gamma_\mu\partial_\nu\psi - \partial_\nu\bar{\psi}\gamma_\mu\psi + \bar{\psi}\gamma_\nu\partial_\mu\psi - \partial_\mu\bar{\psi}\gamma_\nu\psi). \tag{3}$$

In particular, we are interested in the light-cone component,

$$T_{++} = \frac{i}{2}(\psi^\dagger A\partial_+\psi - \partial_+\psi^\dagger A\psi), \tag{4}$$

where we define $A = \gamma_0\gamma_+$.

The decomposition of the fermionic quantum field into Fourier modes yields the following:

$$\psi(x) = \sum_{k,\alpha} b_\alpha(k)u^\alpha(k)e^{-ik\cdot x} + d_\alpha^\dagger(k)v^\alpha(k)e^{ik\cdot x}, \tag{5}$$

where here we are considering discrete Dirac quantization in a box of side $L$. The spinors $u^\alpha(k)$ and $v^\alpha(k)$, where $\alpha = 1, 2$ labels the two independent spin states, are given by

$$u^\alpha(k) = \begin{bmatrix} \sqrt{\frac{\omega_k+m}{2\omega_k V}}\phi^\alpha \\ \frac{\vec{\sigma}\cdot\vec{k}}{\sqrt{2\omega_k(\omega_k+m)V}}\phi^\alpha \end{bmatrix}, \quad \text{and} \quad v^\alpha(k) = \begin{bmatrix} \frac{\vec{\sigma}\cdot\vec{k}}{\sqrt{2\omega_k(\omega_k+m)V}}\phi^\alpha \\ \sqrt{\frac{\omega_k+m}{2\omega_k V}}\phi^\alpha \end{bmatrix}, \tag{6}$$

in which the two dimensional column vectors $\phi^\alpha$ are $\phi^{1\dagger} = (1, 0)$, $\phi^{2\dagger} = (0, 1)$ and $V = L^3$. The normalization has been chosen so that

$$\sum_\alpha ||u^\alpha(k)||^2 = \sum_\alpha ||v^\alpha(k)||^2 = \frac{2}{V}. \tag{7}$$

At the end of the derivation, we will take the continuous limit at $L \to +\infty$.

Now, with these expansions, we can expand the first term of $T_{++}$,

$$\begin{aligned} \frac{i}{2}(\psi^\dagger A\partial_+\psi) = \frac{1}{2}\sum_{k,\tilde{k},\alpha,\alpha'} &\tilde{k}_+ b_\alpha^\dagger(k)b_{\alpha'}(\tilde{k})u_\alpha^\dagger(k)Au_{\alpha'}^\dagger(\tilde{k})e^{i(k-\tilde{k})\cdot x} \\ &- \tilde{k}_+ b_\alpha^\dagger(k)d_{\alpha'}^\dagger(\tilde{k})u_\alpha^\dagger(k)Av_{\alpha'}(\tilde{k})e^{i(k+\tilde{k})\cdot x} \\ &+ \tilde{k}_+ d_\alpha(k)b_{\alpha'}(\tilde{k})v_\alpha^\dagger(k)Au_{\alpha'}^\dagger(\tilde{k})e^{-i(k+\tilde{k})\cdot x} \\ &- \tilde{k}_+ d_\alpha(k)d_{\alpha'}^\dagger(\tilde{k})v_\alpha^\dagger(k)Av_{\alpha'}^\dagger(\tilde{k})e^{-i(k-\tilde{k})\cdot x}, \end{aligned} \tag{8}$$

and similarly for the second term. Normal ordering will switch $d_\alpha(k)$ with $d_{\alpha'}^\dagger(\tilde{k})$ providing an additional minus sign:

$$\begin{aligned} :T_{++}: := \frac{1}{2}\sum_{k,\tilde{k},\alpha,\alpha'} &(k_+ + \tilde{k}_+)[b_\alpha^\dagger(k)b_\alpha(\tilde{k})u_\alpha^\dagger(k)Au_{\alpha'}(\tilde{k})e^{i(k-\tilde{k})\cdot x} \\ &\qquad + d_{\alpha'}^\dagger(\tilde{k})d_\alpha^\dagger(k)v_\alpha^\dagger(k)Av_{\alpha'}(\tilde{k})e^{-i(k-\tilde{k})\cdot x}] \\ &+ (k_+ - \tilde{k}_+)[d_\alpha(k)b_\alpha(\tilde{k})v_\alpha^\dagger(k)Au_{\alpha'}(\tilde{k})e^{-i(k+\tilde{k})\cdot x} \\ &\qquad - b_\alpha^\dagger(k)d_{\alpha'}(\tilde{k})u_\alpha^\dagger(k)Av_{\alpha'}(\tilde{k})e^{i(k+\tilde{k})\cdot x}]. \end{aligned} \tag{9}$$

We are interested in smearing $T_{++}$ in the $x^+$-direction. So, let us put all the other inputs to zero:

$$\begin{aligned} :T_{++}:(x^+, 0) = \frac{1}{2}\sum_{k,\tilde{k},\alpha,\alpha'} &(k_+ + \tilde{k}_+)[b_\alpha^\dagger(k)b_\alpha(\tilde{k})u_\alpha^\dagger(k)Au_{\alpha'}(\tilde{k})e^{i(k_+-\tilde{k}_+)\cdot x^+} \\ &\qquad + d_{\alpha'}^\dagger(\tilde{k})d_\alpha^\dagger(k)v_\alpha^\dagger(k)Av_{\alpha'}(\tilde{k})e^{-i(k_+-\tilde{k}_+)\cdot x^+}] \\ &+ (k_+ - \tilde{k}_+)[d_\alpha(k)b_\alpha(\tilde{k})v_\alpha^\dagger(k)Au_{\alpha'}(\tilde{k})e^{-i(k_++\tilde{k}_+)\cdot x^+} \\ &\qquad - b_\alpha^\dagger(k)d_{\alpha'}(\tilde{k})u_\alpha^\dagger(k)Av_{\alpha'}(\tilde{k})e^{i(k_++\tilde{k}_+)\cdot x^+}]. \end{aligned} \tag{10}$$

For general configurations, the expression above is point-wise unbounded from below, so we have to introduce a smearing. Define, for a smearing function $f$ for which we assume to have the positivity condition $f = g^2$ for some other real function $g$, the smeared energy-momentum tensor component:

$$T_f = \int_{-\infty}^{+\infty} dx^+ : T_{++} : (x^+, 0) f(x^+). \tag{11}$$

By the definition of the fourier transform, $\hat{f}(k) = \int_{-\infty}^{+\infty} dx f(x) e^{-ikx}$, we get the following expression:

$$
\begin{aligned}
T_f = \frac{1}{2} \sum_{k,\tilde{k},\alpha,\alpha'} (k_+ + \tilde{k}_+) &[ b_\alpha^\dagger(k) b_\alpha(\tilde{k}) u_\alpha^\dagger(k) A u_{\alpha'}(\tilde{k}) \hat{f}(\tilde{k}_+ - k_+) \\
&+ d_{\alpha'}^\dagger(\tilde{k}) d_\alpha^\dagger(k) v_\alpha^\dagger(k) A v_{\alpha'}(\tilde{k}) \hat{f}(k_+ - \tilde{k}_+) ] \\
+ (k_+ - \tilde{k}_+) &[ d_\alpha(k) b_\alpha(\tilde{k}) v_\alpha^\dagger(k) A u_{\alpha'}(\tilde{k}) \hat{f}(k_+ + \tilde{k}_+) \\
&- b_\alpha^\dagger(k) d_{\alpha'}(\tilde{k}) u_\alpha^\dagger(k) A v_{\alpha'}(\tilde{k}) \hat{f}(-k_+ - \tilde{k}_+) ].
\end{aligned}
\tag{12}
$$

Note that the matrix $A$ has eigenvalues $\lambda_1 = \lambda_2 = 0$, $\lambda_3 = \lambda_4 = 2$, so it is positive semi-definite. Then it is possible to find the matrix $B$ such that $B^\dagger B = B^\dagger B = A$, i.e. $B$ is the square root matrix of $A$. It is easy to obtain $B$ explicitly but we will only use its existence.

Define the following family of operators for $i \in \{1, 2, 3, 4\}$ and $\mu \in \mathbb{R}$:

$$\mathcal{O}_{\mu i} = \sum_{k,\alpha} \overline{\hat{g}(-k_+ + \mu)} b_\alpha(k) (B u_\alpha(k))_i + \overline{\hat{g}(k_+ + \mu)} d_\alpha^\dagger(k) (B v_\alpha(k))_i, \tag{13}$$

$$\mathcal{O}_{\mu i}^\dagger = \sum_{k,\alpha} \hat{g}(-k_+ + \mu) b_\alpha^\dagger(k) (u_\alpha^\dagger(k) B^\dagger)_i + \hat{g}(k_+ + \mu) d_\alpha(k) (v_\alpha^\dagger(k) B^\dagger)_i. \tag{14}$$

So that $\mathcal{O}_\mu$ for a fixed $\mu \in \mathbb{R}$ is a four-dimensional vector of operators, and $\mathcal{O}_\mu^\dagger$ a co-vector of the same type. Using the anti-commutation relations of the fields, one finds

$$
\begin{aligned}
\mathcal{O}_\mu^\dagger \mathcal{O}_\mu = S_\mu^\nu \mathbb{1} + \sum_{k,\tilde{k},\alpha,\alpha'} &\hat{g}(-k_+ + \mu) \overline{\hat{g}(-\tilde{k}_+ + \mu)} b_\alpha^\dagger(k) b_{\alpha'}(\tilde{k}) u_\alpha^\dagger(k) A u_{\alpha'}(\tilde{k}) \\
&- \hat{g}(k_+ + \mu) \overline{\hat{g}(\tilde{k}_+ + \mu)} d_{\alpha'}^\dagger(k) d_\alpha(\tilde{k}) v_\alpha^\dagger(k) A v_{\alpha'}(\tilde{k}) \\
&+ \hat{g}(k_+ + \mu) \overline{\hat{g}(-\tilde{k}_+ + \mu)} d_\alpha(k) b_{\alpha'}(\tilde{k}) v_\alpha^\dagger(k) A u_{\alpha'}(\tilde{k}) \\
&+ \hat{g}(-k_+ + \mu) \overline{\hat{g}(\tilde{k}_+ + \mu)} b_\alpha^\dagger(k) d_{\alpha'}^\dagger(\tilde{k}) u_\alpha^\dagger(k) A v_{\alpha'}(\tilde{k}),
\end{aligned}
$$

where we have defined

$$S_\mu^\nu \equiv \sum_{k,\alpha} \hat{g}(k_+ + \mu) \overline{\hat{g}(\tilde{k}_+ + \mu)} \delta_{\alpha,\alpha'} \delta_{k,\tilde{k}} v_\alpha^\dagger(k) A v_{\alpha'}(\tilde{k}) = \sum_{k,\alpha} \hat{g}(k_+ + \mu) \overline{\hat{g}(k_+ + \mu)} v_\alpha^\dagger(k) A v_\alpha(k). \tag{15}$$

Proven in the literature [3], the following lemma alows us to recover $T_f$.

**Lemma 1** *Let $f = g^2$ with $g$ a real, smooth, compactly-supported[2] function. Then the following identity holds:*

$$(k_+ + \tilde{k}_+) \hat{f}(k_+ - \tilde{k}_+) = \frac{1}{\pi} \int_{-\infty}^{\infty} d\mu \, \mu \, \hat{g}(k_+ - \mu) \overline{\hat{g}(\tilde{k}_+ - \mu)}.$$

---

[2]Note that the assumption of compact support is stronger than necessary. For example, the lemma still holds for $g$ a Gaussian distribution, since the rapid decay of the function secures convergence of the integral.

Using this lemma,

$$T_f = \frac{1}{2\pi} \int_{-\infty}^{\infty} d\mu \mu (\mathcal{O}_\mu^\dagger \mathcal{O}_\mu - S_\mu^v \mathbb{1}). \tag{16}$$

One can then compute the anti-commutator of the operator $\mathcal{O}$,

$$\begin{aligned}
\{\mathcal{O}_{\mu i}^\dagger, \mathcal{O}_{\mu i}\} &= \sum_{k,\tilde{k},\alpha,\alpha'} (\hat{g}(-k_+ + \mu)\overline{\hat{g}(-\tilde{k}_+ + \mu)}\delta_{\alpha,\alpha'}\delta_{k,\tilde{k}}u_\alpha^\dagger(k)Au_{\alpha'}(\tilde{k}) \\
&\quad + \hat{g}(k_+ + \mu)\overline{\hat{g}(\tilde{k}_+ + \mu)}\delta_{\alpha,\alpha'}\delta_{k,\tilde{k}}v_\alpha^\dagger(k)Av_{\alpha'}(\tilde{k}))\mathbb{1} \\
&= (S_{-\mu}^u + S_\mu^v)\mathbb{1}.
\end{aligned} \tag{17}$$

Note that since $g$ is real valued, $|\hat{g}|$ is even.

Using the anti-commutation relation obtained above, we can split the integral,

$$\begin{aligned}
T_f &= \frac{1}{2\pi} \int_{-\infty}^{\infty} d\mu \mu (\mathcal{O}_{\mu i}^\dagger \mathcal{O}_{\mu i} - S_\mu^v \mathbb{1}) \\
&= \frac{1}{2\pi} \int_0^{\infty} d\mu \mu (\mathcal{O}_{\mu i}^\dagger \mathcal{O}_{\mu i} - S_\mu^v \mathbb{1}) + \frac{1}{2\pi} \int_{-\infty}^0 d\mu \mu (S_{-\mu}^u \mathbb{1} - \mathcal{O}_{\mu i} \mathcal{O}_{\mu i}^\dagger).
\end{aligned} \tag{18}$$

Notice that if $\mu \geq 0$, then $\mu\langle \mathcal{O}_{\mu i}^\dagger \mathcal{O}_{\mu i}\rangle_\psi \geq 0$, and similarly if $\mu \leq 0$, then $-\mu\langle \mathcal{O}_{\mu i}\mathcal{O}_{\mu i}^\dagger\rangle_\psi \geq 0$, for any state $|\psi\rangle$. Hence,

$$\begin{aligned}
\langle T_f \rangle_\psi &\geq -\frac{1}{2\pi} \int_0^{\infty} d\mu \mu S_\mu^v + \frac{1}{2\pi} \int_{-\infty}^0 d\mu \mu S_{-\mu}^u \\
&= -\frac{1}{2\pi} \int_0^{\infty} d\mu \mu (S_\mu^v + S_\mu^u).
\end{aligned} \tag{19}$$

The computation preformed in appendix B shows that

$$\sum_\alpha u_\alpha^\dagger(k)Au_\alpha(k) = \frac{1}{V}\left(1 - \frac{k^1}{\omega_k}\right), \tag{20}$$

$$\sum_\alpha v_\alpha^\dagger(k)Av_\alpha(k) = \frac{1}{V}\left(1 - \frac{k^1}{\omega_k}\right). \tag{21}$$

Finally, plugging it in the expression for $S^u$ and $S^v$ and taking the continuous limit $\frac{1}{V}\sum_{\vec{k}} \to \int \frac{d^3\vec{k}}{(2\pi)^3}$, we come to the conclusion that

$$\langle T_f \rangle_\psi \geq -\frac{1}{\pi} \int_0^{\infty} d\mu \mu \int \frac{d^3 k}{(2\pi)^3} |\hat{g}(k_+ + \mu)|^2 \left(1 - \frac{k^1}{\omega_k}\right). \tag{22}$$

We will denote this bound as $\mathcal{B}_1$, where the subscript 1 represents the number of smearing directions, and the dependence on the smearing function is implicit.

Unfortunately, the integral obtained in equation (22) diverges. By definition, $k_+ = \frac{1}{2}(\omega_k + k_1) = \frac{1}{2}(\omega_k - k^1) = \frac{1}{2}(\sqrt{k_1^2 + k_2^2 + k_3^2 + m^2} - k^1)$, so we can change the integral variable accordingly. Using that the measure transforms as $dk_+ = \frac{1}{2}(\frac{k^1}{\omega_k} - 1)dk^1$, we have that the bound is proportional to

$$\mathcal{B}_1 \propto -\int_0^{\infty} d\mu \int dk_+ \mu |\hat{g}(k_+ + \mu)|^2 \int dk^2 dk^3. \tag{23}$$

We can then note that the integrals in $k_2$ and $k_3$ are decoupled and they will contribute with the volume of the space in those directions. Since the integral in $\mu$ and $k_+$ does not vanish for non-trivial smearing functions, the expression above diverges, which means the bound is completely trivial.

This outcome is clearly unsatisfactory since our aim was to derive a non-trivial lower bound. Such a bound would allow us to explore the extent to which the Null Energy Condition (NEC) is violated within the framework of free fermionic quantum field theory.

However, this divergence is not unexpected, drawing an analogy with the divergence of the bound of the free bosonic theory [10], when the UV cut-off approaches zero. The main issue is that, in order to obtain a convergent integral, it is necessary to fully smear it in the time direction. Note that $t$ is linearly dependent on $x^+$ and $x^-$ due to $x^+ + x^- = t$. Hence, we expect that, if we smear $T_{++}$ in both light-cone directions, we will obtain a convergent lower bound. An earlier treatment can be found in [21], which also gives general reasons for the lack of any quantum energy inequalities over a finite null segment in dimensions higher than 2.

# 3 Double smeared null energy condition

## 3.1 Derivation of the non-trivial bound

In this section, we will prove a convergent lower bound, $\mathcal{B}_2$, by smearing $T_{++}$ in both the $x^+$ and $x^-$-direction. In general, the smearing function can be of the form $f(x^+, x^-)$. For practical purposes, we restrict our argument to the case where $f$ is separable, i.e. the function factors multiplicatively $f(x^+, x^-) = f_+(x^+)f_-(x^-)$. In other words, we are now interested in obtaining a lower bound for

$$T_{f_+ f_-} = \int dx^+ \int dx^- : T_{++} : (x^+, x^-, 0) f_+(x^+) f_-(x^-). \tag{24}$$

Keeping in mind that $e^{ik\cdot x} = e^{ik_+ x^+} e^{ik_- x^-} e^{-ik_\perp \cdot x^\perp}$ and using the definition of Fourier transform, we can carry out the same procedure as before to write

$$T_{f_+ f_-} = \frac{1}{2} \sum_{k,\tilde{k},\alpha,\alpha'} (k_+ + \tilde{k}_+)[b_\alpha^\dagger(k)b_\alpha(\tilde{k})u_\alpha^\dagger(k)Au_{\alpha'}(\tilde{k})\hat{f}_+(\tilde{k}_+ - k_+)\hat{f}_-(\tilde{k}_- - k_-) \tag{25}$$

$$+ d_{\alpha'}^\dagger(\tilde{k})d_\alpha^\dagger(k)v_\alpha^\dagger(k)Av_{\alpha'}(\tilde{k})\hat{f}_+(k_+ - \tilde{k}_+)\hat{f}_-(k_- - \tilde{k}_-)]$$

$$+ (k_+ - \tilde{k}_+)[d_\alpha(k)b_\alpha(\tilde{k})v_\alpha^\dagger(k)Au_{\alpha'}(\tilde{k})\hat{f}_+(k_+ + \tilde{k}_+)\hat{f}_-(k_- + \tilde{k}_-)$$

$$- b_\alpha^\dagger(k)d_{\alpha'}^\dagger(\tilde{k})u_\alpha^\dagger(k)Av_{\alpha'}(\tilde{k})\hat{f}_+(-k_+ - \tilde{k}_+)\hat{f}_-(-k_- - \tilde{k}_-)].$$

Denoting $g_\pm = \sqrt{f_\pm}$, we define the new operators:

$$\mathcal{O}_{\mu i} = \sum_{k,\alpha} \overline{\hat{G}(-k+\mu)} b_\alpha(k)(Bu_\alpha(k))_i + \overline{\hat{G}(k+\mu)} d_\alpha^\dagger(k)(Bv\alpha(k))_i, \tag{26}$$

$$\mathcal{O}_{\mu i}^\dagger = \sum_{k,\alpha} \hat{G}(-k+\mu) b_\alpha^\dagger(k)(u_\alpha^\dagger(k)B^\dagger)_i + \hat{G}(k+\mu) d_\alpha^\dagger(k)(v_\alpha^\dagger(k)B^\dagger)_i, \tag{27}$$

where $\hat{G}(k+\mu) = \hat{g}_+(k_+ + \mu_+)\hat{g}_-(k_- + \mu_-)$ and $\mu_\pm$ are two dummy variables that shall be integrated out in the end. We will denote $\mu = (\mu_+, \mu_-)$.

Since the Fourier transform of the product is the convolution, $f = g^2$ implies that its Fourier transform $\hat{f} = \frac{1}{2\pi}\int d\mu\, \hat{g}(\mu)\hat{g}(k-\mu)$, so we have

$$\hat{f}_-(k_- - \tilde{k}_-) = \frac{1}{2\pi}\int d\mu\, \hat{g}_-(\mu)\overline{\hat{g}_-(\mu - (k_- - \tilde{k}_-))} \tag{28}$$

$$= \frac{1}{2\pi}\int d\mu_-\, \hat{g}_-(k_- - \mu_-)\overline{\hat{g}_-(\tilde{k}_- - \mu_-)}, \tag{29}$$

where we changed the variable $\mu = k_- - \mu_-$ and used that since $g_-$ is real, $\overline{\hat{g}_-(x)} = \hat{g}_-(-x)$. Applying lemma 1 to $f_+$ and $g_+$, one obtains

$$(k_+ + \tilde{k}_+)\hat{f}_+(k_+ - \tilde{k}_+) = \frac{1}{\pi}\int_{-\infty}^{\infty} d\mu_+\, \mu_+ \hat{g}_+(k_+ - \mu_+)\overline{\hat{g}_+(\tilde{k}_+ - \mu_+)}. \tag{30}$$

Then for the double smearing case, using equation 29, we have

$$(k_+ + \tilde{k}_+)\hat{f}_+(k_+ - \tilde{k}_+)\hat{f}_-(k_- - \tilde{k}_-) = \frac{1}{2\pi^2}\int d\mu_+ d\mu_-\, \mu_+ \hat{g}_+(k_+ - \mu_+) \\ \times \overline{\hat{g}_+(\tilde{k}_+ - \mu_+)}\hat{g}_-(k_- - \mu_-)\overline{\hat{g}_-(\tilde{k}_- - \mu_-)}. \tag{31}$$

Taking advantage of the modified lemma and symmetry,

$$\frac{1}{2\pi^2}\int_{-\infty}^{+\infty} d\mu_+ \int_0^{+\infty} d\mu_-\, \mu_+ (\mathcal{O}_\mu^\dagger \mathcal{O}_\mu - S_\mu^v \mathbb{1}) \tag{32}$$

$$= \sum_{k,\tilde{k},\alpha,\alpha'} \frac{1}{2\pi^2}\int_{-\infty}^{+\infty} d\mu_+\, \mu_+ \int_0^{+\infty} d\mu_- \frac{1}{2}(\hat{g}(-k_- + \mu_-)\hat{g}(\tilde{k}_- - \mu_-) + \hat{g}(-k_- - \mu_-)\hat{g}(\tilde{k}_- + \mu_-))$$

$$\times (\hat{g}(-k_+ + \mu_+)\hat{g}(\tilde{k}_+ - \mu_+)b_\alpha^\dagger(k)b_{\alpha'}(\tilde{k})u_\alpha^\dagger(k)Au_{\alpha'}(\tilde{k})$$
$$- \hat{g}(k_+ + \mu_+)\hat{g}(-\tilde{k}_+ - \mu_+)d_{\alpha'}^\dagger(k)d_\alpha(\tilde{k})v_\alpha^\dagger(k)Av_{\alpha'}(\tilde{k}))$$
$$+ \frac{1}{2}(\hat{g}(k_- + \mu_-)\hat{g}(\tilde{k}_- - \mu_-) + \hat{g}(k_- - \mu_-)\hat{g}(\tilde{k}_- + \mu_-))$$
$$\times (\hat{g}(k_+ + \mu_+)\hat{g}(\tilde{k}_+ - \mu_+)d_\alpha(k)b_{\alpha'}(\tilde{k})v_\alpha^\dagger(k)Au_{\alpha'}(\tilde{k})$$
$$+ \hat{g}(-k_+ + \mu_+)\hat{g}(\tilde{k}_+ + \mu_+)b_\alpha^\dagger(k)d_{\alpha'}^\dagger(\tilde{k})u_\alpha^\dagger(k)Av_{\alpha'}(\tilde{k}))$$

$$= \frac{1}{4\pi^2}\sum_{k,\tilde{k},\alpha,\alpha'}\int_{-\infty}^{+\infty} d\mu_+\, \mu_+ \int_{-\infty}^{+\infty} d\mu_- \hat{g}(-k_- + \mu_-)\hat{g}(\tilde{k}_- - \mu_-)$$

$$\times (\hat{g}(-k_+ + \mu_+)\hat{g}(\tilde{k}_+ - \mu_+)b_\alpha^\dagger(k)b_{\alpha'}(\tilde{k})u_\alpha^\dagger(k)Au_{\alpha'}(\tilde{k})$$
$$- \hat{g}(k_+ + \mu_+)\hat{g}(-\tilde{k}_+ - \mu_+)d_{\alpha'}^\dagger(k)d_\alpha(\tilde{k})v_\alpha^\dagger(k)Av_{\alpha'}(\tilde{k}))$$
$$+ \hat{g}(k_- + \mu_-)\hat{g}(\tilde{k}_- - \mu_-)$$
$$\times (\hat{g}(k_+ + \mu_+)\hat{g}(\tilde{k}_+ - \mu_+)d_\alpha(k)b_{\alpha'}(\tilde{k})v_\alpha^\dagger(k)Au_{\alpha'}(\tilde{k})$$
$$+ \hat{g}(-k_+ + \mu_+)\hat{g}(\tilde{k}_+ + \mu_+)b_\alpha^\dagger(k)d_{\alpha'}^\dagger(\tilde{k})u_\alpha^\dagger(k)Av_{\alpha'}(\tilde{k})), \tag{33}$$

where now the definition of $S_\mu^v$ is different from the once-smeared case:

$$S_\mu^v = \sum_{k,\alpha} |\hat{g}_+(k_+ + \mu_+)|^2 |\hat{g}_-(k_- + \mu_-)|^2 v_\alpha^\dagger(k)Av_\alpha(k) \tag{34}$$

$$= \frac{1}{V}\sum_k |\hat{g}_+(k_+ + \mu_+)|^2 |\hat{g}_-(k_- + \mu_-)|^2 \left(1 - \frac{k^1}{\omega_k}\right), \tag{35}$$

and the anti-commutator is what we expect, with the new definitions of $S_\mu^v$ and $S_\mu^u$:

$$\{\mathcal{O}_{\mu i}^\dagger, \mathcal{O}_{\mu i}\} = (S_{-\mu}^u + S_\mu^v)\mathbb{1}. \tag{36}$$

Comparing with the expression of (25), in a analogous way as before, we can prove that

$$\langle T_{f_+ f_-}\rangle \geq -\frac{1}{4\pi^2}\int_0^{+\infty}d\mu_+\int_0^{+\infty}d\mu_-\int\frac{d^3\vec{k}}{(2\pi)^3}\mu_+|\hat{g}_+(k_+ + \mu_+)|^2|\hat{g}_-(k_- + \mu_-)|^2\left(1 - \frac{k^1}{\omega_k}\right).\tag{37}$$

We know that

$$k_+ = \frac{1}{2}(\omega_k + k_1),\tag{38}$$

$$k_- = \frac{1}{2}(\omega_k - k_1),\tag{39}$$

$$\omega_k^2 = k_1^2 + k_2^2 + k_3^2 + m^2.\tag{40}$$

So setting $k_\perp := \sqrt{k_2^2 + k_3^2}$ we obtain,

$$4k_+ k_- = k_2^2 + k_3^2 + m^2 = k_\perp^2 + m^2,\tag{41}$$

and it's straightforward to find that

$$d(k_+ k_-) = \frac{1}{2}k_\perp dk_\perp.\tag{42}$$

Since $dk_1 \wedge dk_2 \wedge dk_3 = dk_1 \wedge dk_\perp \wedge k_\perp d\theta$, where $\theta$ is the angular variable in polar coordinates, we can rewrite part of our integral measure in terms of $dk^+$, $dk^-$ and $d\theta$, i.e. $dk_1 \wedge dk_2 \wedge dk_3 = 2(k_+ + k_-)dk_- \wedge dk_+ \wedge d\theta$.

With all the considerations discussed above, one can change the variable of the integral in the double-smeared bound (37),

$$\langle T_{f_+ f_-}\rangle \geq -\frac{1}{4\pi^2}\int_0^{+\infty}d\mu_+\int_0^{+\infty}d\mu_-\int_{\mathcal{D}}2(k_- + k_+)dk_- dk_+ 2\pi\frac{1}{(2\pi)^3}\mu_+$$

$$\times|\hat{g}_+(k_+ + \mu_+)|^2|\hat{g}_-(k_- + \mu_-)|^2\left(\frac{2k_+}{k_- + k_+}\right)$$

$$= -\frac{1}{4\pi^4}\int_0^{+\infty}d\mu_+\int_0^{+\infty}d\mu_-\int_{\mathcal{D}}dk_+ dk_- \mu_+ k_+|\hat{g}_+(k_+ + \mu_+)|^2|\hat{g}_-(k_- + \mu_-)|^2,\tag{43}$$

where the integration domain is $\mathcal{D} = \{k_\pm \geq 0 | k_+ k_- \geq m^2\}$.

Equation (43) can be further simplified by changing variables. Setting $u = k_+ + \mu_+$ and $v = k_- + \mu_-$,

$$\langle T_{f_+ f_-}\rangle \geq -\frac{1}{4\pi^4}\int_0^{\infty}du\int_{\frac{m^2}{u}}^{\infty}dv\int_{\frac{m^2}{v}}^{u}dk_+\int_{\frac{m^2}{k_+}}^{v}dk_-(u - k_+)k_+|\hat{g}_+(u)|^2|\hat{g}_-(v)|^2.\tag{44}$$

Performing the $k^-$ and $k^+$ integrals, we can present our main result.

$$\boxed{\langle T_{f_+ f_-}\rangle \geq -\frac{1}{4\pi^4}\int_0^{\infty}du\int_{\frac{m^2}{u}}^{\infty}dv\left(\frac{vu^3}{6} - \frac{m^2 u^2}{2} + \frac{m^4 u}{2v} - \frac{m^6}{6v^2}\right)|\hat{g}_+(u)|^2|\hat{g}_-(v)|^2.}\tag{45}$$

It's worth mentioning that the form of our bound looks simpler than the result for the bosonic case in [2]. What appears in our expression is simply an integral of polynomial with smearing function while the result (equation (50)) in [2] involves a parameter $\eta$ remained to be fixed in the parameter space for a minimum value.

The result cannot be further simplified without assuming the exact form of the smearing function. However, we can still analyze whether this integral converges qualitatively. For the first two terms, the absolute value of the integral can be bounded by the same integral but changing the lower integration limit of $v$ to 0. For example,

$$\int_0^\infty du \int_{\frac{m^2}{u}}^\infty dv \frac{vu^3}{6}|\hat{g}_+(u)|^2|\hat{g}_-(v)|^2 \leq \frac{1}{6}\int_0^\infty du\, u^3|\hat{g}_+(u)|^2 \int_0^\infty dv\, v|\hat{g}_-(v)|^2. \tag{46}$$

Since $|\hat{g}_+(u)|^2$ and $|\hat{g}_-(v)|^2$ decay faster than any polynomial when $u$ or $v$ approaches $\infty$, the first two terms of the bound converge. The last two terms in the integrand are coupled to a factor of $v^a$ with $a = -1, -2$ and the following bound holds,

$$\int_{\frac{m^2}{u}}^\infty dv\, v^a|\hat{g}_-(v)|^2 \leq \left(\frac{m^2}{u}\right)^a \int_0^\infty dv\, |\hat{g}_-(v)|^2. \tag{47}$$

The integral on the left-hand side is finite for any $u \geq 0$ so we can be certain that the original integral in the variable $v$ converges, as well as the integral over $u$.

In the next subsections, we will explore this bound in more specific circumstances, where we can find simpler analytic expressions or numerical results.

## 3.2 Massless, Gaussian-smeared bound

Let us first investigate the massless case, where we can compute some analytic results for specific smearing functions. Take the Gaussian function $|\hat{g}_+(u)|^2 = \sigma_+ e^{-(\sigma_+ u)^2}$ (similarly $|\hat{g}_-(v)|^2 = \sigma_- e^{-(\sigma_- v)^2}$)[3] as a particular example of smearing function. By changing variables ($\tilde{u} = \sigma_+ u$ and $\tilde{v} = \sigma_- v$), from equation (45) we obtain the expression for the bound,

$$\langle T_{f_+ f_-} \rangle \geq -\frac{1}{24\pi^4} \int_0^\infty d\tilde{u} \int_0^\infty d\tilde{v} \frac{1}{\sigma_+^3 \sigma_-} \tilde{u}^3 \tilde{v} e^{-\tilde{u}^2} e^{-\tilde{v}^2} \tag{48}$$

$$= -\frac{1}{96\pi^4 \sigma_+^3 \sigma_-}, \tag{49}$$

which turns out to be a satisfactory finite negative number. So, we obtained a non-trivial lower bound for the doubled-smeared $T_{++}$ for the simple case where the smearing is Gaussian. Moreover, $\sigma_+$ has a larger effect on the bound comparatively to $\sigma_-$. This asymmetry of the dependence on the deviations is expected since the energy-momentum tensor component considered has, by definition, a preferred spacetime direction.

Since large $\sigma_\pm$ correspond to a wide smearing in spacetime, we expect the bound to approach zero. This is indeed in agreement with the well-studied averaged null energy condition (ANEC) [22]. On the other hand, in the $\sigma_\pm \to 0$ case, i.e. there is no smearing in spacetime, we obtain a trivial bound. This is expected since the expected value of the energy-momentum evaluated at a particular spacetime point is generally unbounded.

Comparing with the result of [2] in the massless limit, we find that our fermionic result shares similar features with the bosonic case. Taking $n = 4$ and the limit of $m \to 0$, equation (1) in [2] turns into

$$\langle T_{--}^{\text{smear}} \rangle \geq -\frac{\mathcal{N}_2}{(\delta^+)(\delta^-)^3}, \tag{50}$$

---

[3]We are defining our Gaussian function slightly different from the usual form $\frac{1}{\sqrt{2\pi}\sigma}e^{-\frac{1}{2}(\frac{u}{\sigma})^2}$ here. This way, $\sigma_\pm$ are the smearing lengths in spacetime. It's convenient for the later comparison with the previous result because the variables for our smearing function in the final expression are the wave number after the Fourier transform.

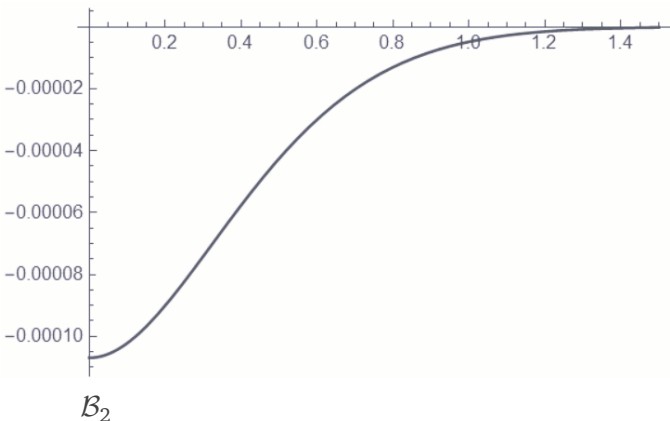

Figure 1: The bound $\mathcal{B}_2$ as a function of the mass $m$.

where $\mathcal{N}_2$ is simply a constant since $m \to 0$. Now that we are evaluating $T_{++}^{\text{smear}}$ instead of $T_{--}^{\text{smear}}$, the role of $\delta^+$ and $\delta^-$ will exchange as expected and our pre-factor $\frac{1}{48\pi^4}$ takes place of the previous constant $\mathcal{N}_2$.

As a consistency check, let us take $\sigma_- \to 0$ while keeping $\sigma_+$ fixed. In this limit, the right-hand side of equation (49) diverges to $-\infty$, once again yielding a trivial bound when smearing in a single direction.

## 3.3 Mass dependence of the Gaussian-smeared bound

One can also wonder about how this bound, which will now denote by $\mathcal{B}_2$, depends on the mass. Let us choose the two smearing functions to be Gaussians with standard deviation $\sigma = 1$, i.e. $|\hat{g}_+(x)|^2 = |\hat{g}_-(x)|^2 = e^{-x^2}$. By dimensional analysis, we have $[\sigma]=-1$. Now the bound takes the following form:

$$\mathcal{B}_2 = -\frac{1}{4\pi^4} \int_0^\infty du \int_{\frac{m^2}{u}}^\infty dv \left( \frac{vu^3}{6} - \frac{m^2 u^2}{2} + \frac{m^4 u}{2v} - \frac{m^6}{6v^2} \right) e^{-(u^2+v^2)}. \tag{51}$$

We can numerically integrate the expression above to obtain the following relation between $\mathcal{B}_2$ and the mass, which is shown in Figure 1. Since we work with natural units, $u$ and $v$ are in the same unit as $m$, so $[\mathcal{B}_2]= 4$. It's worth mentioning that the features of Figure 1 are close to the blue line in Figure 1 of [2], which again shows the similarity between fermionic and bosonic case.

Note that in the highly massive region, the lower bound approaches zero, which also appears in the bosonic case [2]. We can understand this result in the following qualitative way. Roughly speaking, quantum effects are relevant when the de Broglie wavelength of the particle, $(\frac{h}{mv})$, is much greater than the characteristic size of the system, $d$. In our case, we simply take this $d$ to be the smearing length. For small $m$, quantum behavior becomes prominent, but as $m$ increases, classical behavior dominates. Given that the classical case satisfies the Null Energy Condition (NEC), the bound is anticipated to approach zero as $m$ becomes large, which is verified numerically in the figure above.

## 4 Conclusion

In this work, we investigated the (Double) Smeared Null Energy Condition for the fermionic free theory in 4-dimensional flat Minkowski spacetime.

We first obtained an inequality for the once-smeared $T_{++}$. This inequality holds for all physically reasonable states. Actually, a fully rigorous formulation would require it to hold for all Hadamard states, as the claim in [3], since we are using the similar procedure. However, unlike the result in [3], we illustrated that our derived lower bound diverges. The reason for this is that instead of smearing $T_{00}$ directly in time direction in [3], time is not fully smeared in our case when we only smear in direction $x_+$ for $T_{++}$.

We addressed the triviality of this once-smeared $T_{++}$ later by applying the smearing in two directions, providing a new energy condition. We offered explicit analytic results for the massless case and numerical insights for the mass-dependence of the latter bound in the case of Gaussian smearing.

Regarding the outlook of this research, as mentioned in [1], understanding the behavior of DSNEC in interacting field theories remains still an open question. Though we are using different methods than those used in [1] to derive the DSNEC for fermions, our approach still specifically relies on the canonical quantization of the Dirac field and the form of the basis wavefunctions satisfying the Dirac equation. It remains to be further investigated whether the stress-tensor can be exactly expressed by terms involving an operator of the type $\mathcal{O}\mathcal{O}^\dagger$ in an interacting theory.

Moreover, our discussion is limited to Minkowski spacetime. Extending the DSNEC to curved spaces is a highly significant direction, given that it is crucial for its application in semiclassical gravity. It would then be interesting to explore a generalized version of our results in different curved spacetimes.

## Acknowledgments

We thank Tiago Scholten for the stimulating discussions, and Professor Ben Freivogel for the helpful supervision. We also thank the reviewers for their valuable feedback, which helped strengthen this article.

**Funding information** We also thank the support of the summer internship program organized by the D&I councils of IoP (UvA) and P&A (VU).

## A  Conventions

In this paper, we work in 4 dimensional Minkowski spacetime with the "mostly minus" signature in natural units ($c = \hbar = 1$).

The position light-cone coordinates are given by $x^\pm = t \pm x^1$. The corresponding metric tensor for the coordinates $(x^+, x^-, x^2, x^3)$ is,

$$g_{\mu\nu} = \begin{pmatrix} 0 & 1/2 & 0 & 0 \\ 1/2 & 0 & 0 & 0 \\ 0 & 0 & -1 & 0 \\ 0 & 0 & 0 & -1 \end{pmatrix}. \tag{A.1}$$

In momentum space we will denote $k_\pm = \frac{1}{2}(\omega_k \pm k_1)$ such that $k \cdot x = k_+ x^+ + k_- x^- + x_i \cdot k^i$.

In our convention, the 4-by-4 gamma matrices in the position space are defined as

$$\gamma^0 = \begin{pmatrix} 1 & 0 & 0 & 0 \\ 0 & 1 & 0 & 0 \\ 0 & 0 & -1 & 0 \\ 0 & 0 & 0 & -1 \end{pmatrix}, \tag{A.2}$$

$$\gamma^i = \begin{pmatrix} 0 & \sigma_i \\ -\sigma_i & 0 \end{pmatrix}. \tag{A.3}$$

Since we are working with the mostly-minus metric we obtain

$$\gamma_0 = \gamma^0, \tag{A.4}$$

$$\gamma_i = -\gamma^i. \tag{A.5}$$

In particular,

$$\gamma_+ = g_{+-}\gamma^- = \frac{1}{2}(\gamma^0 - \gamma^1) = \frac{1}{2}\begin{pmatrix} 1 & 0 & 0 & -1 \\ 0 & 1 & -1 & 0 \\ 0 & 1 & -1 & 0 \\ 1 & 0 & 0 & -1 \end{pmatrix}, \tag{A.6}$$

and we define

$$A = \gamma_0\gamma_+ = \frac{1}{2}\begin{pmatrix} 1 & 0 & 0 & -1 \\ 0 & 1 & -1 & 0 \\ 0 & -1 & 1 & 0 \\ -1 & 0 & 0 & 1 \end{pmatrix} = \frac{1}{2}(\mathbb{1} - \mathbb{L}), \tag{A.7}$$

where $\mathbb{1}$ is the identity matrix, and $\mathbb{L}$ is the exchange matrix.

# B   Explicit computations for $\sum_\alpha u_\alpha^\dagger(k)Au_\alpha(k)$

In Appendix B, the explicit computations for $\sum_\alpha u_\alpha^\dagger(k)Au_\alpha(k)$ will be made.

Note that $\sum_\alpha u_\alpha^\dagger(k)Au_\alpha(k) = \frac{1}{V} - \frac{1}{2}\sum_\alpha u_\alpha^\dagger(k)\mathbb{L}u_\alpha(k)$, using the decomposition $A = \frac{1}{2}(\mathbb{1} - \mathbb{L})$ and the normalization of $u_\alpha(k)$. Then, we can write

$$u_1(k) = \begin{bmatrix} a_1 \\ C b_1 \end{bmatrix}, \tag{B.1}$$

where we define

$$a_1 = \sqrt{\frac{\omega_k + m}{2\omega_k V}}\begin{bmatrix} 1 \\ 0 \end{bmatrix}, \tag{B.2}$$

$$b_1 = \frac{1}{\sqrt{2\omega_k(\omega_k + m)V}}\begin{bmatrix} 1 \\ 0 \end{bmatrix}, \tag{B.3}$$

$$C = \vec{\sigma} \cdot \vec{k}. \tag{B.4}$$

Using this notation we obtain that

$$u_1^\dagger(k)\mathbb{L}u_1(k) = a_1^\dagger\sigma_1 C b_1 + b_1^\dagger C^\dagger\sigma_1 a_1. \tag{B.5}$$

The matrix $\sigma_1$ appears in the non-zero blocks of $\mathbb{L}$.

Now,

$$
\begin{aligned}
a_1^\dagger \sigma_1 C b_1 &= \sqrt{\frac{\omega_k + m}{2\omega_k V}} \begin{bmatrix} 1 & 0 \end{bmatrix} k^1 \frac{1}{\sqrt{2\omega_k(\omega_k + m)V}} \mathbb{1}_{2\times 2} \begin{bmatrix} 1 \\ 0 \end{bmatrix} \\
&\quad + \sqrt{\frac{\omega_k + m}{2\omega_k V}} \begin{bmatrix} 1 & 0 \end{bmatrix} k^2 \frac{1}{\sqrt{2\omega_k(\omega_k + m)V}} \begin{bmatrix} i & 0 \\ 0 & -i \end{bmatrix} \begin{bmatrix} 1 \\ 0 \end{bmatrix} \\
&\quad + \sqrt{\frac{\omega_k + m}{2\omega_k V}} \begin{bmatrix} 1 & 0 \end{bmatrix} k^3 \frac{1}{\sqrt{2\omega_k(\omega_k + m)V}} \begin{bmatrix} 0 & -1 \\ 1 & 0 \end{bmatrix} \begin{bmatrix} 1 \\ 0 \end{bmatrix} \quad \text{(B.6)}
\end{aligned}
$$

$$
= \frac{1}{V} \frac{1}{2\omega_k}(k^1 + ik^2). \tag{B.7}
$$

For the second term, we have something similar:

$$
\begin{aligned}
a_1^\dagger \sigma_1 C b_1 &= \sqrt{\frac{\omega_k + m}{2\omega_k V}} \begin{bmatrix} 1 & 0 \end{bmatrix} k^1 \frac{1}{\sqrt{2\omega_k(\omega_k + m)V}} \mathbb{1}_{2\times 2} \begin{bmatrix} 1 \\ 0 \end{bmatrix} \\
&\quad + \sqrt{\frac{\omega_k + m}{2\omega_k V}} \begin{bmatrix} 1 & 0 \end{bmatrix} k^2 \frac{1}{\sqrt{2\omega_k(\omega_k + m)V}} \begin{bmatrix} -i & 0 \\ 0 & i \end{bmatrix} \begin{bmatrix} 1 \\ 0 \end{bmatrix} \\
&\quad + \sqrt{\frac{\omega_k + m}{2\omega_k V}} \begin{bmatrix} 1 & 0 \end{bmatrix} k^3 \frac{1}{\sqrt{2\omega_k(\omega_k + m)V}} \begin{bmatrix} 0 & 1 \\ -1 & 0 \end{bmatrix} \begin{bmatrix} 1 \\ 0 \end{bmatrix} \quad \text{(B.8)}
\end{aligned}
$$

$$
= \frac{1}{V} \frac{1}{2\omega_k}(k^1 - ik^2). \tag{B.9}
$$

With this we conclude that

$$
u_1^\dagger(k) \mathbb{L} u_1(k) = \frac{1}{V} \frac{1}{\omega_k} k^1. \tag{B.10}
$$

The analogous computation for $u_2(k)$ yields the same result. For $u_2(k)$,

$$
u_2^\dagger(k) \mathbb{L} u_2(k) = \frac{1}{V} \frac{1}{\omega_k} k^1. \tag{B.11}
$$

Summing both, we obtain:

$$
u_\alpha(k)^\dagger(k) \mathbb{L} u_\alpha(k) = \frac{2}{V} \frac{1}{\omega_k} k^1. \tag{B.12}
$$

In the same way, we can obtain the exact same result for $v_\alpha(k)^\dagger(k) \mathbb{L} v_\alpha(k)$.

In conclusion,

$$
\sum_\alpha u_\alpha^\dagger(k) A u_\alpha(k) = \frac{1}{V}\left(1 - \frac{k^1}{\omega_k}\right), \tag{B.13}
$$

$$
\sum_\alpha v_\alpha^\dagger(k) A v_\alpha(k) = \frac{1}{V}\left(1 - \frac{k^1}{\omega_k}\right). \tag{B.14}
$$

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
