# Peer review of "The fermionic double smeared null energy condition"

_SciPost Physics Core, doi:SciPost Phys. Core 8, 045 (2025)_

## Round 1 · Referee Report · Anonymous (Referee 1) · 2024-12-10

Strengths

1- This is the first work to consider the DSNEC for non-scalar fields.
2- Generally clear presentation.

Weaknesses

1- Some typos and possible missing factors.

Report

Pointwise energy conditions are not respected in quantum field theory, but a variety of quantum energy inequalities have been shown to hold under suitable circumstances. This nicely written paper derives a double-smeared null energy condition DSNEC for Dirac fields in Minkowski space, adapting a method developed by Fewster and Mistry [3] for averages along timelike curves to the double-smeared setting. DSNEC for scalar fields was derived in previous work [1,2].

Apart from a few typos, the derivation appears correct and after some minor corrections I would recommend it for publication.

Requested changes

1- In the abstract both refs [1] & [2] are cited as sources for DSNEC, but in the introduction only [1] is. I think that [2] should also appear here, because this is where DSNEC was actually proved, as far as I understand it.
2- After (5), various symbols, e.g. u's, v's, b's and d's should be defined.
3- After (7), "smear" should be "smearing"
4- In (11) the \alpha subscript is misplaced in the final term
5- In (12) there should be no dagger on the d_\alpha
6- Above (26) I believe that the convolution integral should include a 1/(2\pi) factor and the authors should check the remaining calculation for any consequences.
Also the g's should be \hat{g}'s
7- Similarly (29) lacks factors of \pi and 2\pi
8- After (39) the variable \theta should be explained.
9- After (42) there should be some commentary on why this integral converges, due to the rapid decay of the transforms and the integration range.
10- After (44) sigma_1,2 should be replaced by \sigma_\pm
11- I suggest that the authors also consider the limit of transverse smearing tends to zero, without changing \sigma_+, for consistency check with Section 2
12- In (51) there appears to be a missing factor of 1/2 [cf (35)]. Again, the authors should check for any consequences.
13- In the reference list, the capitalisation of proper names, e.g. Dirac, and acronyms like ANEC, QNEC should be corrected.
14- The authors note that smearing along a null geodesic without transverse smearing results in a trivial bound. They correctly state on p.5 that this is expected, citing the divergence of the bound obtained in [10] as a UV cutoff is removed. An earlier treatment can be found in Phys. Rev. D 67 (2003) 044003, which also gives general reasons for the lack of any QEI over a finite null segment in dimensions higher than 2.
15- A remark: In their conclusion, the authors say that it would be interesting to generalise their results to curved spacetimes. I expect that this is possible, and a starting point might be Class. Quantum Grav. 23 (2006) 6659-6681, which generalises [3] to globally hyperbolic spacetimes.

Recommendation

Ask for minor revision

---

## Round 1 · Referee Report · Anonymous (Referee 2) · 2024-12-27

Strengths

1 - Establishes new lower bounds on smeared energy density which may play a helpful role in geometric theorems for semi-classical.
2 - Well written and explained with clear technical manipulations.

Weaknesses

1 - Results are a generalization and match the expectations of previous literature. Can be given more context with the previous literature.
2 - Unclear how this can be extended beyond free fermions (as authors claim).
3 - Several typos appearing in equations and text.

Report

In this article the authors consider the ‘smeared null energy’, i.e. the null components of the stress tensor integrated against a smearing function, of free Dirac fields in four-dimension Minkowski space. Adapting the general schematic of Fewster and Mistry (citation [3] in the article), they express the smeared null stress-tensor as a purely positive operator minus a state-independent functional of the smearing functions. This lower bounds the expectation value of the smeared null energy in all states of the theory. This leads to two results:

1) Smearing over only one lightcone direction leads to a divergent lower bound, suggesting the potential for this operator to be unbounded in the free Dirac theory. This fits with similar results established for free bosonic fields.

2) Following the procedure established by Fliss, Freivogel, and Kontou (citations [1,2] in the article), they consider the ‘double smeared null energy’ in which the null stress-tensor is integrated against smearing functions in both lightcone directions. They show this integral is finite and establishes a new, meaningful, lower bound for null energy densities in the free Dirac theory (what they call the ‘fermionic DSNEC’). The authors establish how this bound depends on the mass of the Dirac field and illustrate (through numerical integration) how the bound tightens in the large mass limit.

The results of this article are not necessarily surprising (given the literature of smeared null energy densities for free bosonic theories and the results for the ‘Quantum Weak Energy Inequality’ in [3]) and the technical aspects of the computation follow closely [3]. Regardless, the paper is well written, and clearly explained. To the extent that I have worked through the technical calculations, the results of the article are correct and I believe are a useful addition to the literature on energy conditions in quantum field theories. For this reason I tentatively recommend the article for publication in SciPost Physics subject to some medium and small revisions.

Requested changes

Medium revisions (aimed to try to establish some context for the main results of the article):

1) The authors illustrate that their derived lower bound diverges when smeared over a single lightcone direction, however this does not quite establish that the smeared null energy is an unbounded operator in the theory (i.e. potentially there could exist a tighter, more well behaved bound). Can the authors construct example states where the smeared null energy is not bounded below? The reason for this question is that the example state of [1] for the corresponding free bosonic theory is a squeezed state: this state is a superposition over all particle numbers localized to roughly the same transverse momentum. Such superpositions are not possible in a fermionic theory due to exclusion so it is not clear that the same sort of unbounded negative null energy density will arise. It would be nice and would solidify the author’s first result if they can illustrate explicit states with unbounded smeared null energy.

2) Using Gaussian smearing functions, the authors illustrate the dependence of the fermionic DSNEC on the characteristic smearing length scales (e.g. in eq. (44) for the massless theory) which takes the same form as expected from the free bosonic theory in [2]. It would be helpful if the authors could make comparison to the results to [2] (say to establish whether they are the same or if one is tighter over the other) in both the massless and massive cases.

3)In the Conclusion the authors state “... our approach may be promising for interacting field theories. This is because our techniques are not as specifically tailored to treat free field theories.” I am confused by this statement because it seems much of the technical results rely on the canonical quantization of the Dirac field and the form of the basis wavefunctions satisfying the Dirac equation. For example, in an interacting theory it is not clear what the right definition of normal ordering might be or if the stress-tensor can be exactly expressed as a perfect square (minus a state independent constant). Can the authors please elaborate more concretely on this claim?

Minor revisions related to typos, potential misclaims, or grammatical errors that the authors should address:
1) In equation (6) there is a missing dagger on the d_{\alpha’} in the second line. This error propagates through equations (7), (8), and (10). Regardless, this error is corrected in the writing of T as a perfect square in equation (16) and so does not affect the result. The same error occurs again however in equation (23).
2) In equation (11) the \alpha should appear as a superscript on v^\alpha.
3) In equation (12) there should not be a \dagger on the the d^\dagger_\alpha.
4) In the sentence above equation (22) the authors state “we are now interested in obtaining an upper bound for…” I am certain they actually mean “a lower bound”.
5) In the last paragraph in section 3.2 the authors state “This is indeed in agreement with the well-studied null energy condition (NEC) [21].” Do the authors mean the averaged null energy condition (ANEC)? The usual null energy condition is a pointwise condition (i.e. \sigma_\pm ->0 as opposed to \infinity) and is known to be unbounded in QFT (as pointed out by the authors in the two subsequent sentences).
6) Footnote 3 is incomplete.
7) In appendix B, the authors have given the explicit form for u_1(k). It would then also be helpful to list u_2, and v_{1,2} for readers wishing to reproduce their calculations.

Recommendation

Ask for minor revision

---

## Round 2 · Referee Report · Anonymous (Referee 1) · 2025-5-27

Strengths
1- This is the first work to consider the DSNEC for non-scalar fields. 2- Generally clear presentation.
Weaknesses
1- Some typos and possible missing factors.
Report
a) In Appendix A, as noted in my first report, the expression for $ \gamma_+$ in (A.6) is missing a factor of 1/2, arising from lowering the index from $\gamma^+ =\gamma^0+\gamma^1$ using the metric (A.1). Compare with the factor of 1/2 in $k_+$. This should be corrected and has potential consequences throughout the manuscript, as it feeds in to the matrix A and the stress-energy tensor component $T_{++}$.
b) On p.7, l.127, "applying lemma 1 again" is incorrect - it is (29) that is used here.
c) l.152-153 - it's not quite correct to say that all terms vanish, because that is not true of the last one. The discussion of convergence could be simplified by estimating each term in the $v$-integral separately. The $v$-integral for terms with a power $v^a$ for $a\ge 0$ can be estimated less than $\int_0^\infty dv\, v^a |\hat{g}(v)|^2$ multiplied by terms depending on $u$, while those with $a<0$ can be estimated less than $(m^2/u)^a \int_0^\infty dv |\hat{g}(v)|^2$, again multiplied by terms depending on $u$. The $v$-integrals now converge as do the remaining u ones.
Regarding the journal - it seems to me that this is not ground-breaking in the sense required for SciPost Physics but it is in principle publishable in SciPost PhysicsCore.
Requested changes
1- Fix the factor in $\gamma_+$ and any consequences throughout the manuscript 2- Fix wording as in comment (b) 3- Consider improving the wording as in (c).
Recommendation
Ask for minor revision

---

## Round 2 · Author Response

My co-author and I were pleased to receive your response and feedback inviting us to revise and resubmit our manuscript. Accordingly, we would like to submit the enclosed revised paper, for reconsideration for publication as we have revised our manuscript according to the reviewers’ comments.
We would like to thank you for providing your constructive and detailed review comments on our manuscript. The recommendations and advice have helped us to enhance the quality of the manuscript.
All authors have read and approved the revised manuscript. We hope that our resubmission is now suitable for inclusion in SciPost and we look forward to hearing from you.
Thank you for your time,
Best regards,
Duarte Fragoso

---

## Round 2 · List of Changes

We added some remarks regarding the divergence of single smeared bound both in section 2 and in the conclusion.
We added a comparison to the results in reference [2] in both the massless and massive cases.
We elaborated on the claim regarding the outlook of generalizing the fermionic DSNEC to interacting theory.
To clarify the context,
After (5), various symbols, e.g. u's, v's, b's and d's are be defined.
After (43) the variable \theta is explained.
After (46) we comment on why this integral converges
We add a consistency check after equation (49)
We wrote the proof for double smearing case more explicitly
We also correct several typos pointed out by the reviewers.
The missing dagger in equation (8), (9), (10), (12) and (25) on the d_{\alpha’}
The \alpha appearing as a superscript In equation (13) on v^\alpha.
In equation (14) there should not be a \dagger on the the d^\dagger_\alpha.
In the sentence above equation (24) it’s “a lower bound” rather than “an upper bound”
In the last paragraph in section 3.2 we mean the averaged null energy condition (ANEC) instead of NEC
Complete Footnote 3.
After (9), we change “smear" into “smearing"
The missing 1/(2\pi) factor in the convolution integral and the pre-factor involved
After (48) sigma_1,2 is replaced by \sigma_\pm
The capitalisation of proper names in the reference list.
Moreover, we thanked the reviewers in the aknowlegments.

---

## Round 3 · Author Response

Dear reviewer.

Thank you for the detailed comments regarding the convergence of our bounds and other useful feedback.
On my and my co-author's behalf, I am resubmitting the paper, after correcting the typos and inaccuracies pointed out in the review.

Thank you again for the help,

Best regards,

Duarte Fragoso

---

## Round 3 · List of Changes

We fixed the missing factor of 1/2 in the expression of \gamma_+ (and the consequences of this overall factor throughout the paper)

On page 7, line 127 we changed "lemma 1" to the correct equation (equation 29)

We changed the argument regarding convergence of the bound given in equation 46 by analyzing the positive and negative powers of v separately, as suggested by the reviewer, bounding the original integrals by others that are evidently finite.

---

## Editorial Decision

published